# The genetic basis for the adaptation of *E. coli* to sugar synthesis from CO$_2$

Elad Herz[1], Niv Antonovsky[1], Yinon Bar-On[1], Dan Davidi[1], Shmuel Gleizer[1], Noam Prywes[1], Lianet Noda-Garcia[2], Keren Lyn Frisch[1], Yehudit Zohar[1], David G. Wernick[1], Alon Savidor[3], Uri Barenholz [1] & Ron Milo [1]

Understanding the evolution of a new metabolic capability in full mechanistic detail is challenging, as causative mutations may be masked by non-essential "hitchhiking" mutations accumulated during the evolutionary trajectory. We have previously used adaptive laboratory evolution of a rationally engineered ancestor to generate an *Escherichia coli* strain able to utilize CO$_2$ fixation for sugar synthesis. Here, we reveal the genetic basis underlying this metabolic transition. Five mutations are sufficient to enable robust growth when a non-native Calvin–Benson–Bassham cycle provides all the sugar-derived metabolic building blocks. These mutations are found either in enzymes that affect the efflux of intermediates from the autocatalytic CO$_2$ fixation cycle toward biomass (*prs*, *serA*, and *pgi*), or in key regulators of carbon metabolism (*crp* and *ppsR*). Using suppressor analysis, we show that a decrease in catalytic capacity is a common feature of all mutations found in enzymes. These findings highlight the enzymatic constraints that are essential to the metabolic stability of auto-catalytic cycles and are relevant to future efforts in constructing non-native carbon fixation pathways.

[1] Department of Plant and Environmental Sciences, Weizmann Institute of Science, Rehovot 7610001, Israel. [2] Department of Biomolecular Sciences, Weizmann Institute of Science, Rehovot 7610001, Israel. [3] de Botton Institute for Protein Profiling, The Nancy and Stephen Grand Israel National Center for Personalized Medicine, Weizmann Institute of Science, Rehovot, 7610001, Israel. Elad Herz and Niv Antonovsky contributed equally to this work. Correspondence and requests for materials should be addressed to R.M. (email: ron.milo@weizmann.ac.il)

The Calvin–Benson–Bassham (CBB) cycle[1] is the primary pathway producing organic biomass from inorganic carbon, accounting for an overwhelming majority of global primary biomass production[2]. Due to the pivotal role of the CBB cycle in the carbon balance of the biosphere, significant efforts have been devoted to enhance $CO_2$ fixation in autotrophic organisms, especially in crops, to boost biomass accumulation rates. Despite the rapid expansion of molecular tools for genetic engineering in autotrophs such as plants and algae, these organisms are significantly less malleable for genetic manipulation in comparison to non-photosynthetic model organisms, such as *Escherichia coli*.

Recently, we showed that a combination of rational metabolic engineering and laboratory evolution can achieve the successful integration of a non-native CBB cycle into *E. coli*[3]. The evolved strain is able to utilize $CO_2$ as the sole carbon source for the synthesis of all sugar-derived metabolic building blocks (e.g., ribose-5-phosphate), while cellular energy and additional metabolites are obtained from an externally supplied organic carbon source (e.g., pyruvate). The successful evolution of a hemiautotrophic strain was the first proof-of-concept demonstration that a non-native carbon fixation pathway can be integrated into the metabolic network of a heterotrophic host to allow autocatalytic conversion of $CO_2$ to biomass.

Notably, our initial strain design, based on rational rewiring of the metabolic network of the host using gene deletions and insertions, failed to sustain sugar production from $CO_2$ even though all the required enzymatic machinery was heterologously expressed. To facilitate the successful integration of the synthetic pathway, the engineered ancestral strain was subjected to continuous laboratory evolution that selected for mutants in which $CO_2$ fixation was able to sustain sugar synthesis. Indeed, a few weeks of laboratory evolution under intense selection in a xylose-limited chemostat led to the emergence of evolved strains (termed hemiautotrophic), in which all sugar was synthesized from $CO_2$[3].

In the previous study[3], whole-genome sequencing of three parallel evolutionary experiments revealed that each of the hemiautotrophic strains that took over the population contained 20–40 mutations, in addition to the rationally designed genetic modifications that were introduced to the ancestor strain. We then identified genes that were repeatedly mutated in these independent evolutionary experiments and were therefore expected to play a key role in the adaptation process. We introduced two such mutations (*prs* and *ppsR* as described below) into the ancestor strain and tested its ability to grow hemiautotrophically. While the modified ancestor strain that contained these two additional mutations was still unable to grow hemiautotrophically, evolving it in the lab resulted in a hemiautotrophic strain with much fewer required mutations (a total of eight mutations)[3].

The failure of the rationally designed ancestor strain to sustain hemiautotrophic growth in the absence of novel mutations indicated that additional tuning of the metabolic network was required. Which of the mutations that accumulated in the evolved strains are essential for $CO_2$ fixation and what is their function? Genomic drift and "hitchhiker" mutations[4,5] make it challenging to differentiate between causative mutations and non-essential ones. Here, we identify five mutations that are sufficient for the adaptation of the rationally designed ancestor *E. coli* to produce sugar from $CO_2$. Our results elucidate the molecular basis of shortcomings in the original design and suggest the functional role of the mutations that enable stable autocatalytic operation of the non-native CBB cycle.

## Results

### Five extra mutations are sufficient for sugar synthesis from $CO_2$.
To identify which of the mutations that were fixed during

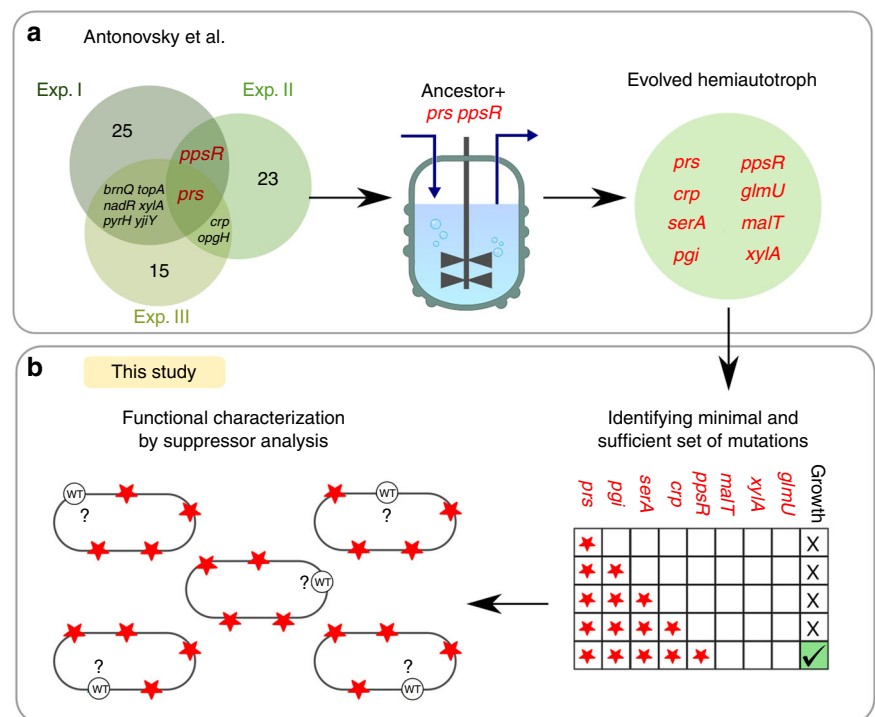

**Fig. 1** Finding a compact sufficient set of mutations from the evolved allele pools. **a** Hemiautotrophic growth, in which $CO_2$ is fixed for sugar synthesis, occurred independently in three chemostat evolutions (green circles, numbers indicate the number of extra mutations not written). Integration of the *prs* and *ppsR* mutations into ancestral strain and "replaying" evolution yielded a much more compact set of mutations (light green) (Antonovsky et al.)[3]. **b** The evolved alleles were combinatorially introduced to find a sufficient set which is locally minimal, i.e., removal of any mutation from the set will cause loss of the phenotype. The reconstructed strain was evaluated and characterized by suppressor analysis to understand the functional role of each mutation

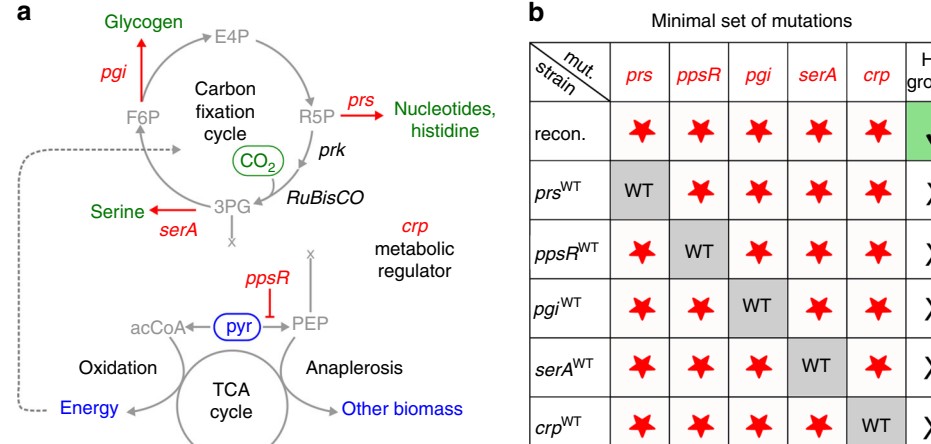

**Fig. 2** A locally minimal set of mutations enables hemiautotrophic growth. **a** Role of mutated genes (red) in central carbon metabolism. Three mutations are located in branch points of the synthetic CBB cycle (*serA*;H210Q, *pgi*;G378C, and *prs*;R105_A110dup). Two mutations are regulators influencing the carbon metabolism (*crp*;M190K and Δ*ppsR*). The initial strain design decouples central carbon metabolism in order to create two distinct modules, one for energy production and second for $CO_2$ fixation. In green, biomass components originating from $CO_2$; in blue, end products of pyruvate supply. Ovals mark the feed substrates. **b** The reconstructed strain has five essential mutations that enable $CO_2$ fixation. Reverting every mutation independently back to WT prevents hemiautotrophic growth on liquid media, showing that this set of mutations is sufficient and irreducible

the laboratory evolution are required for hemiautotrophic growth, we chose to start our investigation with the evolved hemiautotrophic strain that contains mutations in eight genes, namely *prs*, *pgi*, *serA*, *glmU*, *crp*, *ppsR*, *xylA*, and *malT* (Fig. 1). This strain was obtained following ≈50 days of chemostat evolution from the modified ancestor strain that contained mutation in *prs* and *ppsR*, in addition to the rationally designed genetic modifications[3]. To identify the mutations which are essential for the hemiautotrophic phenotype, we used iterative P1 transduction (Supplementary Fig. 1) to introduce subsets of mutations into the genetic background of an unevolved ancestor. The resulting strains, each containing a subset of one or more mutations (Fig. 1 and Supplementary Fig. 1), were transformed with a plasmid containing RuBisCO, *prkA*, and carbonic anhydrase (pCBB) and tested for hemiautotrophic growth by inoculating in minimal media supplemented only with pyruvate in an elevated $CO_2$ atmosphere. Under these conditions, all biosynthetic sugar demands must be met through $CO_2$ fixation[3]. Using our combinatorial reconstruction approach (Supplementary Fig. 1), we identified a subset of mutations, that when introduced into the rationally designed ancestor strain, were sufficient to reproduce hemiautotrophic growth on liquid media. Specifically, as shown in Fig. 2a, this subset was composed of five mutations in the following genes: phosphoglucose isomerase (*pgi*:G378C), 3-phosphoglycerate dehydrogenase (*serA*:H210Q), ribose-phosphate diphosphokinase (tandem duplication of six amino acids, *prs*:R105_A110dup), cAMP receptor protein (*crp*:M190K), and PEP synthetase regulatory protein (*ppsR*:knockout). Next, we constructed five "leave-one-out" strains, each containing only four out of these five mutations. Each strain was tested for its ability to sustain hemiautotrophic growth in liquid. We found that all five leave-one-out strains failed to grow hemiautotrophically under the selective condition, similar to the ancestor strain, indicating that all five mutations are required for reproducing the phenotype (Fig. 2b).

The reconstructed strain, in which hemiautotrophic growth was achieved following the introduction of these five mutations, has a growth rate indistinguishable from that of the evolved hemiautotrophic strains (Supplementary Fig. 2). Interestingly, while the *ppsR* mutation was necessary for robust hemiautotrophic growth in liquid media, it was not required for colony

formation on solid agar plates. More detailed investigation of the *ppsR* leave-one-out strain (that contains the native *ppsR* sequence along the other four mutations) revealed that a small number of inoculated cultures (which depends on the inoculum size, and in our case was ≈10%) were able to grow hemiautotrophically, always with a much slower growth rate and a low final yield. As we discuss below, we interpret the *ppsR* mutation as a refinement mutation that ensures robust growth.

To test if the minimal reconstructed strain fully reproduces the hemiautotrophic phenotype we performed isotopic labeling experiments. We incubated the minimally reconstructed strain in the presence of $^{13}CO_2$ and analyzed the isotopic labeling pattern of proteinogenic amino acids using liquid chromatography–mass spectrometry. As shown in Fig. 3a, the labeling pattern of the minimally reconstructed strain matches the labeling of the evolved strain and demonstrates that all sugar-derived biomass is synthesized from $CO_2$. Isotopic labeling of the total biomass content (Fig. 3b) indicates that sugar-derived biomass building blocks (≈30% of all cellular carbon) originate from $CO_2$ fixation. Cumulatively, the phenotypic and metabolic characterization indicates that the reconstructed strain fully reproduces the hemiautotrophic phenotype that was achieved through directed evolution.

We define this set of five mutations as "locally minimal" as none of the mutations can be removed without being replaced by another mutation in order to reproduce the phenotype. That is, the mutations set cannot be made smaller by using only a subset of these mutations. We note that other locally minimal sets of mutations with little overlap with this set[3] can give rise to the hemiautotrophic phenotype. In the following section, we describe an experimental approach to better understand the functional effect of the mutations in this specific and highly compact locally minimal set by generating locally minimal sets which are overlapping in all but one mutation.

Three of the mutated genes in this locally minimal set encode enzymes that divert intermediates of the autocatalytic $CO_2$ fixation cycle toward biosynthetic routes for biomass production (*pgi*, *serA*, and *prs*, shown in Fig. 2a). Prs, which phosphorylates ribose-5-phosphate, shunts the CBB cycle intermediate ribose-5-phosphate toward the synthesis of 5-phospho-α-D-ribose 1-diphosphate (PRPP), a key component in the biosynthesis of

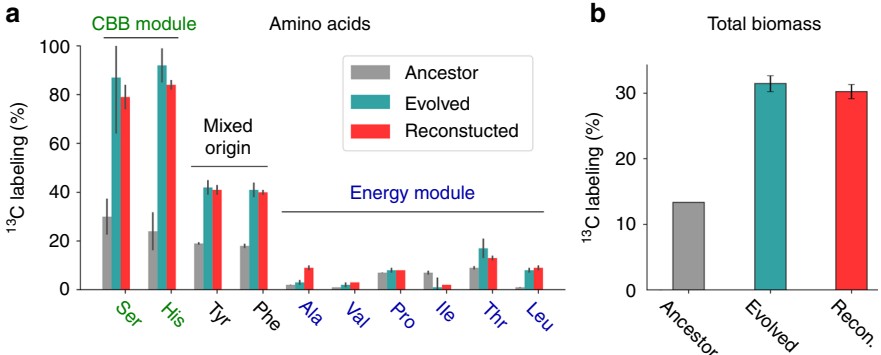

**Fig. 3** Validation of the hemiautotrophic phenotype in the reconstructed strain. **a** We grew the reconstructed, evolved, and ancestral clones with isotopically labeled $^{13}CO_2$ as an inorganic carbon source and non-labeled pyruvate as an energy source, and performed isotopic labeling analysis. Amino acids derived from the CBB cycle (e.g., serine) are labeled green, whereas amino acids derived from the energy module (e.g., alanine) are in blue. As predicted for a hemiautotrophic growth mode, amino acids derived from the intermediates of the CBB module are almost fully labeled in the reconstructed and evolved strain but not in the ancestor strain. In marked contrast, amino acids derived from the TCA and lower glycolysis (energy module) show low levels of labeling. This validates that the reconstructed and evolved strains synthesize CBB module biomass precursors from $CO_2$ using the non-native CBB cycle. **b** We propagated cells in minimal media supplemented with non-labeled pyruvate (5 g/L) and labeled $^{13}CO_2$ atmosphere (0.1 atm). We used an elemental mass analyzer to measure the isotopic carbon composition in the total cell biomass (mean ± SD; $n = 4$). The reconstructed strain shows one-third of cellular carbon originated from fixed inorganic carbon, a value similar to the evolved strains and in line with the expected fraction of biomass precursors originating from the CBB module. The ancestor strain required the addition of xylose (non-labeled, 0.2 g/L) for growth. Labeled carbon in the ancestor strain results from the RuBisCO-dependent catabolism of xylose and from the assimilation of inorganic carbon via anaplerotic reactions

nucleotides. Pgi catalyzes the conversion of the cycle intermediate fructose-6-phosphate to glucose-6-phosphate, as it is needed for producing glycogen through ADP-glucose. SerA catalyzes the first committed step in the biosynthesis of L-serine from the cycle intermediate 3-phosphoglycerate. The two other mutated proteins (Fig. 2a), PpsR and Crp, are both regulators of enzymes in central carbon metabolism. PpsR (PEP synthetase regulatory protein) catalyzes both the $P_i$-dependent activation and ADP/ATP-dependent inactivation of PEP synthetase (PpsA[6]). Crp (cAMP receptor protein) is a transcriptional dual regulator that regulates the expression of ≈400 metabolic and other genes in *E. coli*[7]. Crp is known for the regulation of carbon catabolite repression (CCR)[8, 9] but also coordinates catabolism and anabolism in different nutrient environments[10].

**Mutations' functions investigated via leave-one-out suppressor analysis**. To elucidate the functional role of each mutation, we take advantage of suppressor screens to generate locally minimal sets of mutations that reproduce the phenotype. To this end, we utilized the five leave-one-out strains described in Fig. 2b, each containing only four out of the five essential mutations. For each strain, we performed leave-one-out suppressor analysis that is akin to the classic suppressor screen, but implemented it using deep sequencing and for each of the evolved mutations independently. Briefly, we first cultured a large population (~ $10^{10}$ cells) from each of the leave-one-out strains under relaxed conditions, in which the ability to synthesize sugar from $CO_2$ is not essential for growth (see Methods). Due to the natural error rate during DNA replication, such a large population is likely to contain nearly all possible point mutants[11]. We then selected for the hemiautotrophic phenotype by growth in restrictive conditions in which sugar synthesis from $CO_2$ is essential. As shown in Fig. 4a, the analysis of clones capable of hemiautotrophic growth revealed independent mutants that compensate for the missing mutation. Multiple independent cultures were plated for each leave-one-out strain. We sequenced the hemiautotrophic colonies to identify the compensating mutations and used their annotation to gain insight into the function of each of the mutations which compose the locally minimal set (Fig. 4b–f).

The leave-one-out suppressor analysis enables us to obtain a spectrum of mutations that are expected to play a metabolic role analogous to the original mutation which is absent from the leave-one-out strain. We hypothesized that we would be able to rationalize the function of the original mutations by comparison to the set of compensating mutations that is uncovered in this assay. For example, compensating mutations upstream or downstream in the same pathway demonstrate that modulating the pathway is the essential function. Similarly, a frameshift mutation that replaces a point mutation will suggest that loss of function is required. We proceed to describe the compensating mutations that were identified for each of the mutations in the locally minimal set.

**Knockout of *pgi* can replace the various *pgi* mutations**. To test the function of the *pgi* mutation, we performed leave-one-out suppressor analysis as described above, constructing a strain containing all necessary mutations except *pgi*. We cultured the leave-one-out *pgi*^WT strain under permissive conditions to obtain ~ $10^{10}$ cells and screened them for hemiautotrophic growth (see Methods). We sampled 10 independent clones that were able to grow hemiautotrophically for further analysis using Sanger sequencing of the *pgi* locus or whole-genome sequencing. As shown in Fig. 4b, we found 4 new mutations in the *pgi* coding sequence that reproduced the hemiautotrophic phenotype. In addition, we also identified 6 hemiautotrophic clones in which the *pgi* sequence has not been mutated. Whole-genome sequencing of these clones identified mutations in phosphoglucomutase (*pgm*), a gene downstream of *pgi* in the glycogenesis pathway. Notably, both genes take part in a pathway linking F6P, an intermediate of the CBB cycle, to biomass products such as glycogen. The new mutations observed in *pgi* contained loss-of-function frameshift mutations (Supplementary Table 1). *pgi* is a main sink of CBB cycle intermediates toward G6P, an essential precursor for the formation of biomass components. *E. coli* can grow on gluconeogenic carbon sources in the absence of *pgi*, which suggests that alternative pathways can compensate for its function[12, 13]. To verify that the observed frameshift mutations to *pgi* resulted in the complete loss of enzymatic activity, we showed that

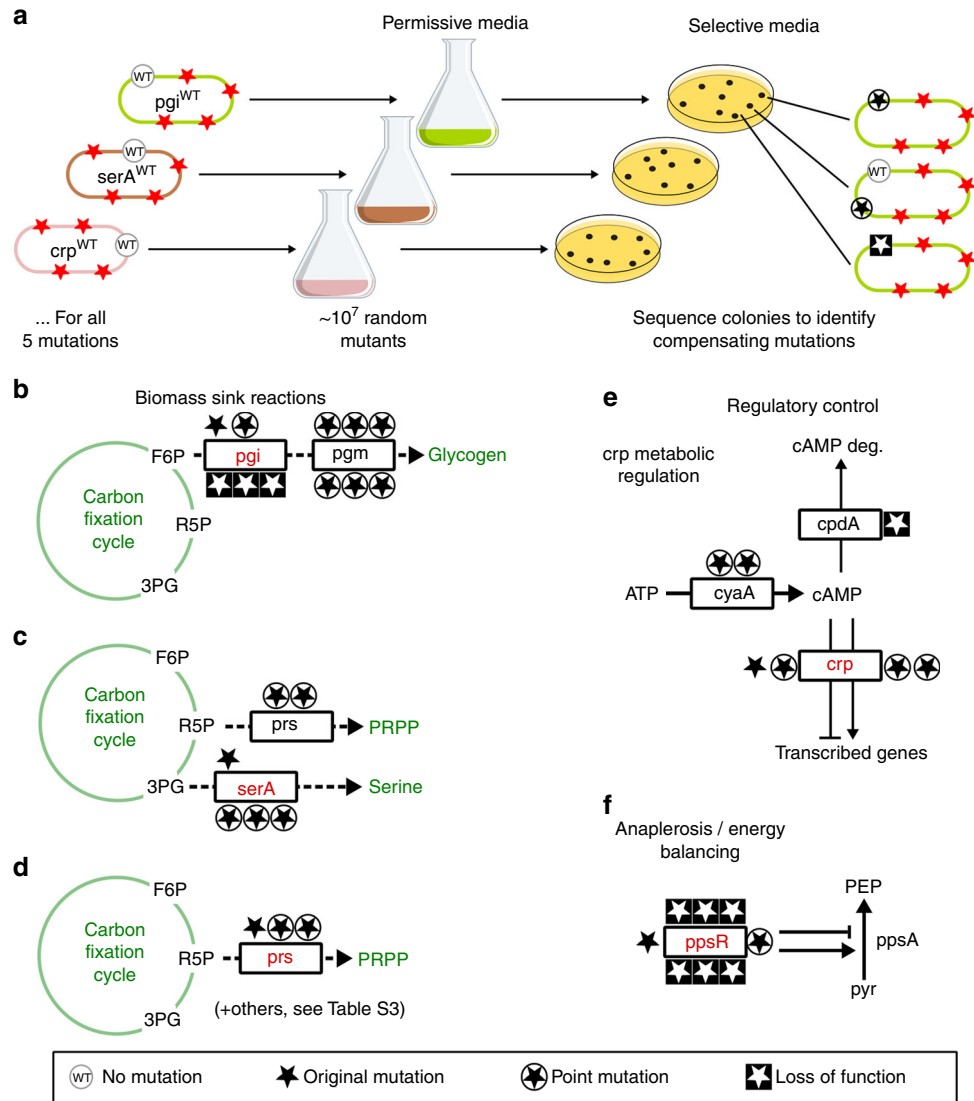

**Fig. 4** Leave-one-out suppressor analysis reveals the functionality of mutations in the minimal set. **a** The reconstructed strain has five essential mutations that enable sugar synthesis from $CO_2$ using the non-native CBB cycle. We constructed "leave-one-out" strains, each missing one essential mutation from the minimal set of mutations that reproduce the hemiautotrophic phenotype, and grew them in permissive conditions to give a population in which many natural mutational variants exist due to DNA replication errors. The population was plated in selective conditions, where CBB activity is essential for growth (Methods), to reveal mutations that regenerate the hemiautotrophic phenotype. Colonies from the plate were sequenced to detect the compensating mutation(s). Several independent cultures ($n > 4$) were cultured and plated for each leave-one-out strain, and multiple colonies were sequenced from each plate. For the sake of graphical simplicity, a single plate is presented. **b** The compensating mutations to *pgi* occurred either in the *pgi* gene itself or in *pgm*, a downstream reaction to biomass (glycogen) production. Most mutations in the *pgi* gene are loss-of-function mutations. **c** Mutations in *serA* were located in the functional regions of the enzyme (e.g., active site and NAD-binding site). We also found that mutations in *prs* can serve as an alternative to the mutation in *serA*. **d** In the *prs* leave-one-out strain we found that compensating mutations in *prs* were located in the functional regions of the enzyme. Other compensating mutations are described in Supplementary Table 3. **e** For the *crp*[WT] strain, most compensating mutations occurred in cAMP biosynthesis and degradation, or in the *crp* gene itself. **f** *ppsR*[WT] managed to grow without further compensating mutations on solid plates, but mutations in *ppsR* took over the population very rapidly in liquid. The wild-type allele introduced in the leave-one-out assay are labeled red. Stars mark unique mutations. Full details for all mutations are listed in Supplementary Tables 1–5

introducing a *pgi* knockout leads to regaining the hemi-autotrophic phenotype (Supplementary Fig. 3).

The catalytic capacity of a reaction, often denoted as $V_{max}$, is defined as $E \cdot k_{cat}$, where $E$ is the amount of the catalyzing enzyme and $k_{cat}$ is the turnover rate of the enzyme[14]. Because $k_{cat}$ is a constant, the catalytic capacity of an enzyme is proportional to the amount of enzyme expressed. As multiple enzymes are capable to shunt F6P toward biosynthesis pathways, the generalized catalytic capacity of the efflux reaction can be defined as the sum of the catalytic capacity of Pgi and the catalytic capacity

resulting from promiscuous activity of other enzymes. Loss-of-function mutation in *pgi* is expected to tune down the efflux capacity of the intermediate F6P from the CBB cycle toward G6P synthesis. This is consistent with other lines of evidence for the required fine-tuning for the operation of the autocatalytic cycle[15] as described in the discussion.

**Mutations in *serA* reduce enzyme catalytic capacity and activity.** Suppressor analysis of a *serA*[WT] strain yielded multiple

compensating mutations, most of which were in the *serA* coding sequence, as shown in Fig. 4c. These mutations affect residues near the active site or at the NAD-binding domain (Supplementary Table 2), and thus could modify the kinetic properties of SerA. Determining the effect of these mutations on the activity of the enzyme using in vitro assays is technically challenging as the equilibrium constant of the forward reaction is very small ($K_{eq} = 8E - 07$, $\Delta_r G'^0 \approx 35 kJ/mol$)[16] and the substrate of the reverse reaction is highly reactive and unstable. Instead, we quantified the rate of 2-oxoglutarate reduction, a known promiscuous activity of *serA*[17–19]. As shown in Supplementary Fig. 4, all SerA mutants presented a 10–100-fold lower in vitro activity compared to the wild-type enzyme. As the decrease in the reaction rate measured for the non-native substrate does not directly indicate a decrease in the reaction rate for the native substrate (e.g., when the mutation affects substrate specificity), we conducted an additional experiment to investigate the effect of the mutations on the native activity of the enzyme in vivo. To this end, we used a wild-type strain in which *serA* has been knocked out and complemented it with plasmids carrying one of the *serA* mutants recovered in the leave-one-out assay. Since SerA is an essential enzyme for growth on minimal media in the absence of serine (or related compounds such as glycine)[20], the growth rate of the strains expressing the mutated SerA is expected to be correlated with the flux carried by the enzyme. Hence, by measuring the growth rate of these strains, we can probe the effect of each mutation on the potential metabolic capacity of *serA* in vivo. As shown in Supplementary Fig. 5, strains complemented with a plasmid-born copy of mutated *serA* enzymes show a decrease in growth rate of 2-fold or more compared to the wild type. The observed decrease in growth rate between the native and mutant *serA* strains is significantly lower than the fold change of the in vitro activity (potentially due to high expression from the plasmid or the different substrate involved). We conclude that all identified mutations share a common outcome: the capacity of 3PG efflux from the Calvin–Benson–Bassham cycle toward biosynthesis is reduced.

Interestingly, in addition to mutations in *serA*, we also found that a second mutation in the *prs* gene can reproduce hemiautotrophic growth on the background of a *serA*$^{WT}$ strain. In one case, the mutation was located in the flexible active site loop (where the original *prs* mutation is located) as an extension of the duplication mutation previously observed. In a different case, the second mutation occurred in an intergenic site upstream to *prs*. As Prs also acts as an enzyme at a flux branch point (as we discuss below), these results suggest an ability to compensate for a needed change at one flux branch point by a change in another flux branch point in the same pathway, in line with a recent analysis[15]. Besides the mutants described above, we detected one more mutant with a *dnaE* mutation (α-subunit of polIII). The role of this mutation is unknown and awaits further investigation.

### Prs reaction rate is reduced by mutations.

*prs* was the only gene that was mutated in all three independent chemostat experiments[3]. Our suppressor analysis of *prs* revealed additional mutations in *prs* that can reproduce the hemiautotrophic phenotype, as detailed in Fig. 4d and Supplementary Table 3. We previously characterized the mutations that were observed in the *prs* gene (ribose-phosphate diphosphokinase)[3] and found that in vitro reaction rates of these prs mutants toward its substrate ribose-5-phosphate showed an ≈2-fold decrease in ($k_{cat}/K_M$) compared with the kinetic parameters of the native enzyme. This includes the R105_A110 duplication mutation contained in the reconstructed strain. In addition, we found that other mutations can reproduce the hemiautotrophic phenotype. Specifically, we

found that mutations in *cpxA*, *proA*, *yjiY*, *pitA*, and *typA* managed to reproduce the phenotype in several independent *prs*$^{WT}$ leave-one-out strains (Supplementary Table 3). We speculate that following the introduction of the mutations in *serA* and *pgi*, which affect carbon branching from the CBB cycle, a mutation in *prs* is no longer required to ensure the autocatalytic stability of the cycle. This is in contrast to the essentiality of the *prs* mutation in the genetic background lacking mutations in *serA* and *pgi*, as was previously observed[3]. It is not clear at this point how mutations that do not seem to be relevant to carbon balancing (e.g., *cpxA*, *proA*, and *gpp*) can compensate to achieve hemiautotrophic growth in the *prs* leave-one-out background. Future work with better experimental resolution will be needed to further investigate this issue and suggest a role for these alternative mutations that compensate for the *prs* mutation in our leave-one-out test.

### Suppressor mutations on a crp$^{WT}$ background support a need for increasing its activity.

In addition to the mutations found in enzymes at metabolic branch points, two main regulators of metabolism were found to be essential: *crp* and *ppsR* . Because a global transcriptional regulator such as *Crp* affects hundreds of genes, it is challenging to infer how a specific mutation in its coding sequence contributes to the emergence of a novel phenotype. For example, point mutations in a transcription factor can lead to over- or underregulation and affect all or a specific subgroup of regulated targets. Therefore, the ability to generate additional mutations that reproduce the phenotype can help us to infer the functional role of the mutation in the context of the hemiautotrophic growth.

Suppressor analysis of the leave-one-out *crp*$^{WT}$ strain revealed that mutations in either the cyclic-AMP (cAMP) biosynthesis or its degradation pathway can reproduce the hemiautotrophic phenotype, as shown in Fig. 4e. We observed several point mutations in CyaA (adenylate cyclase) which catalyzes the synthesis of cAMP. We also identified a loss-of-function mutation in *cpdA* (cAMP phosphodiesterase) which degrades cAMP. Taken together, this indicates that higher cAMP levels and higher Crp activity are selected for hemiautotrophic growth. It was previously reported[21, 22] that Crp activity is increased in the Δ*cpdA* strain therefore influencing the level of transcription of genes regulated by cAMP-Crp. To verify that a *cpdA* loss-of-function mutation reproduces the hemiautotrophic phenotype, we knocked out the *cpdA* gene under the background of *crp*$^{WT}$. Deleting *cpdA* indeed enabled the cells to regain hemiautotrophic growth (Supplementary Fig. 6).

A small number of clones contained mutations not directly related to cAMP: we observed mutations in either *lgoR* or *arcA*, which are global transcription regulators. Two other clones were mutated in more than one locus, as detailed in Supplementary Table 4. These results suggest new connections to the well-studied regulatory role of *crp*, which will require further investigation beyond the scope of this study.

### Refinement mutations in ppsR improve the growth rate and yield.

The *ppsR*$^{WT}$ (Fig. 4a) strain was the only leave-one-out strain able to form colonies on agar plates in restrictive conditions without compensatory mutations (see Methods). However, when colonies were inoculated in liquid media, most inoculums did not exhibit growth. About 10% of the cases, usually with a large inoculum, showed slow growth with very low yield (Supplementary Fig. 7). In these cultures, whole-genome sequencing did not reveal additional mutations that could explain the observed growth. We speculate that the sporadic growth (in the absence of additional mutations) can result from a stochastic transition between metabolic states that occurs only in a small subfraction

of the population[23] or due to a mutation that we could not detect in our usage of next-generation sequencing.

We hypothesize that *ppsR* is a refinement mutation—a mutation which is not strictly essential for growth but improves growth rate and yield significantly. According to this hypothesis, the strain with the *ppsR* native allele is genetically unstable and will rapidly acquire a *ppsR* mutation during bacterial propagation. We demonstrated this by taking seven independent colonies with the WT *ppsR* allele that managed to grow slowly in liquid, and diluted them into fresh pyruvate medium. After overnight growth, the culture was sequenced at the *ppsR* gene and we found that all the cultures had mutations in *ppsR* (Fig. 4f and Supplementary Table 5). Most mutations were transposon insertions and the rest were small deletions or duplications that disrupt the coding sequence. After the mutation was fixed in the population, the culture growth rate was increased to the same value as the reconstructed hemiautotrophic strain.

To further understand the function of the *ppsR* deletion, we performed targeted proteomics analyzing the phosphorylation state of the regulated gene *ppsA*. PpsA is the target of the bifunctional kinase PpsR and it is known that the phosphorylated form of PpsA is inactive[6, 24]. We tested several mutants from our previous evolution experiments[3], as well as evolved *ppsR* knock-out strain compared to the ancestor strain containing wild-type *ppsR*. The data reveal more than two orders of magnitude decrease in the non-active form of PpsA in all the tested mutants (Supplementary Fig. 8), suggesting that elevated activity of PpsA is selected for in our strains.

## Discussion

In this study, we constructed a minimal set of only five mutations that confers to *E. coli* the ability to synthesize its sugars via inorganic carbon fixation. We unravel the identity of these five "fine-tuning" mutations from the larger set of mutations incorporated in the evolutionary process. By generating multiple locally minimal sets of mutations that reproduce the phenotype, we were able to elucidate the functional role of essential mutations. We can divide these mutations into two main types: flux branch reactions out of the autocatalytic cycle and regulators of metabolism. A shared design principle of the mutations related to flux branch points out of the CBB cycle is reduction of the catalytic capacity of the flux branch point reactions. In all three biomass branch points (*pgi*, *serA*, and *prs*), in which intermediates are drained from the CBB cycle toward biomass synthesis, we observe mutations that are expected to decrease the catalytic capacity of the efflux reaction. This is in line with theoretical predictions based on stability analysis of autocatalytic pathways like the CBB cycle[15].

In addition to mutations that fine-tune the metabolic efflux of the autocatalytic carbon fixation pathway, we identify two mutations in regulatory proteins that are essential for the hemi-autotrophic phenotype in the context of a locally minimal set. The first regulatory mutation affects the coding sequence of *crp*. Using leave-one-out suppressor analysis, we demonstrate that a loss-of-function mutation in cAMP-degrading enzyme (*cpdA*) reproduces the phenotype. As *cpdA* knockout results in a two-fold increase in the cellular concentration of cAMP[25], an activator of Crp, we hypothesize that overactivation of Crp is required to sustain hemiautotrophic growth. While further analysis is required to elucidate the exact metabolic mechanism by which Crp activation promotes hemiautotrophic growth, it has been shown that Crp activity regulates the balance between the influx of carbon backbones from catabolism and the efflux of these backbones to anabolism[10]. In our ancestor strain, central carbon metabolism is severed into two modules that supply carbon

backbones independently from one another. We hypothesize that the perturbed supply of carbon backbones shifts the normal balance between catabolism and anabolism to an unstable state, and requires a modified regulation (or modified Crp activity) to restore the balance and enable growth. The second regulatory mutation is a loss of function of *ppsR*, a protein that represses the enzymatic activity of PEP synthetase (*ppsA*). Our phosphopro-teomic analysis (Supplementary Fig. 8), shows that the loss of function of *ppsR* results in a decrease in the phosphorylation state of PpsA that is known to be associated with an increase in *ppsA* activity[6]. Natively, *ppsA* activity is regulated to prevent a futile cycle via several mechanisms whose identity is not fully char-acterized (e.g., allosteric and post-translational). We hypothesize that the establishment of a functional CBB cycle strongly inhibits PpsA activity, e.g., via an increase in the concentrations of CBB cycle intermediates like FBP which is suggested to repress PpsA both allosterically and transcriptionally[26]. In this scenario, loss of function of PpsR to increase the PpsA activity is therefore required to supply PEP from pyruvate for cellular biosynthesis needs. Our suggestion that the *ppsR* mutation solves a futile cycle can explain why there is haphazard slow growth in liquid media that is soon superseded by a refinement mutation in *ppsR*. Overall, by unraveling the roles of "fine-tuning" mutations, we uncovered principles for successful metabolic design, principles that are not included in current analysis tools like stoichiometric flux balance models. Notably, some of the five mutations in the minimally local set do not appear within the three evolution repeats (Fig. 1a) that led to the emergence of the hemiautotrophic strains. We interpret this as another indication of the importance of contingency in evolution and the existence of many potential parallel paths. While some of the original mutations were hitchhikers, others may have been beneficial in the specific genomic context and conditions prevailing in the chemostat but are not essential for the final phenotype.

Our previous[3] and current study demonstrate how to achieve a locally minimal set of mutations that results in a complex evo-lutionary transition. We first repeated the process of evolution several times independently in order to reveal shared targets that can be incorporated into the ancestral strain and serve as a new initial condition for lab evolution. Laboratory evolution starting from these improved initial conditions resulted in a much smaller set of mutations on which combinatorial reconstruction and verification of a locally minimal set becomes feasible. Once a locally minimal set is identified, leave-one-out suppressor analysis can then be used to gain functional understanding as we demonstrate here. Iterative use of leave-one-out suppressor ana-lysis can rapidly generate multiple locally minimal genotypes that exhibit the target phenotype, thus significantly facilitating a mechanistic understanding.

Our results demonstrate how even a dramatic metabolic shift from heterotrophy to carbon fixation and sugar synthesis can occur via a relatively small set of mutations. This highlights the plasticity of metabolism and is reminiscent of the adoption of RuBisCO from heterotrophic metabolism in the distant past[27]—one of the major evolutionary transitions in the history of life. In addition, our study serves as a detailed case study for guiding future synthetic biology efforts at rewiring metabolism and understanding the results of lab evolution.

## Methods

**Growth conditions**. Strains were cultured in M9 minimal media, supplemented with 34 mg/l chloramphenicol and suitable carbon sources, as specified in the text. Ultrapure agarose (Hispanagar) was used for the preparation of solid media. Hemiautotrophic clones were cultured in a shaking incubator (37° C, 220 RPM) with an elevated $CO_2$ atmosphere ($pCO_2 = 0.1$ atm). To recover clones from frozen glycerol stocks, cells were first streaked on solid minimal media supplemented with pyruvate (0.5%) and glycerol (0.1%) to facilitate initial growth. Once growth was

observed, liquid minimal medium supplemented only with pyruvate was inoculated.

**Genomic reconstruction.** The ancestral and evolved strains, as well as the pCBB plasmid are described in detail in ref. [3]. We note that all the genetic manipulations were made in a background of *E. coli* strain BW25113 (Keio collection[28], supplied by Clone and Strain Repository unit, WIS). The mutational reconstruction was accomplished by P1 transduction. P1 lysate was prepared from Keio collection[28] strains containing a Kanamycin (Km) resistance marker in an adjacent locus to the desired allele (as specified in Supplementary Table 6). In the case of *ppsR*, the mutations found in the evolved hemiautotrophic strains were loss-of-function mutations. As such, to introduce the *ppsR* mutation, we transduced the *ppsR*:Km marker into the reconstructed strain using *ppsR* (JW1693) as the donor. Next, strains with the desired mutated allele were transduced with resistance marker P1 phages and plated on Km plates. Colonies containing the desired mutated allele and Km resistance marker were selected as donor strains. A P1 lysate of each donor strain was used for transferring the desired mutations to a recipient strain. Due to the large size (≈100 kb) of the DNA fragment delivered to the recipient, in addition to the selection marker, adjacent genes are often included in the transduction event. Genotyping was performed by multiple allele-specific colony PCR[29](MASC) or Sanger sequencing. For further genomic modification, pCP20 (Clone and Strain Repository unit, WIS), a temperature-sensitive plasmid encoding the FLP recombinase, was used according to standard procedures[30] to eliminate the Km resistance markers, allowing iterative rounds of gene editing. After recombinase expression, the loss of resistance markers was validated using PCR, and the resulting strain was used as the recipient in the next iteration of gene deletion. In the final iteration of the reconstruction process, the genomic kanamycin resistance marker has not been removed. Whole-genome sequencing was used to validate the genotype at the end of the process.

**Genotyping by multiplex allele-specific colony PCR (MASC-PCR).** As described in ref. [29], MASC-PCR enables the detection of single-nucleotide polymorphisms in multiple loci simultaneously. We designed primers for genotyping 5 loci in the evolved genome (Supplementary Table 7). Each locus has a different amplicon size: *serA* (169 bp), *pgi* (267 bp), *crp* (379 bp), *malT* (443 bp), and *xylA* (550 bp). For each targeted locus, three primers are designed: (i) a forward primer specific to the wild-type (F-WT) sequence, (ii) a forward primer specific to the mutant (F-MUT), and (iii) a reverse primer (REV) common to both (Supplementary Table 7). The WT and MUT primer mixes final concentration 5 mM were mixed with KAPA2G Fast Multiplex and ddw. PCR reaction was as the following: initial heating 95c (3 min), then 23 cycles of denaturation 95c (15 s), annealing 71c (30 s), and elongation 72c (40 s). For complete gel separation, we use 2% agar run time of 40 min at 120 V.

**Whole-genome sequencing.** DNA was extracted from sampled cultures using DNeasy Blood & Tissue kit (QIAGEN, Germany), and library preparation was performed as described in ref. [31]. Tagging and fragmenting ('tagmentation') using the Nextera kit (Illumina kits FC-121–1031) was performed by mixing 1 μl containing 1.5 ng of genomic DNA, 1.25 μl of TD buffer, and 0.25 μl of TDE1. The mixture was mixed gently by pipetting and placed for incubation in a thermocycler for 8 min at 55 °C. Next, "tagmented" gDNA underwent PCR-mediated adapter addition and library amplification by mixing 11 μl of PCR master mix (KAPA KK2611/KK2612), 4.5 μl of 5 μM index1 (Nextera index kit FC-121-1011), 4.5 μl of 5 μM index2, and 2.5 μl of tagmented DNA in each well. The final total volume per well was 22.5 μl. The thermocycler was run with the following program: 1) 72 °C for 3 min, 2) 98 °C for 5 min, 3) 98 °C for 10 s, 4) 63 °C for 30 s, and 5) 72 °C for 30 s. 6) Repeat steps (3)–(5) 13 times for a total of 13 cycles. 7) 72 °C for 5 min. 8) Hold at 4 °C. PCR cleanup and size selection was done in several steps: mixing 12 μl of magnetic beads spriSelect reagent (Beckman Coulter B23317) with 15 μl of each PCR reaction. Incubation at room temperature for 5 min followed by 1 min on a magnetic stand. The clear solution was discarded, and the beads were mixed with 200 μl of freshly made 80% ethanol. An ethanol wash was performed twice, and the plate was then incubated at room temperature for 5 min to allow for the evaporation of residual ethanol. The sample was eluted with 30 μl of ultrapure water for 5 min at room temperature, and the beads were removed using the magnetic stand. In some cases, after testing the library size in tapeStation (Agilent 4200), another step of 0.6X "right side selection" was performed to remove large amplicons. The prepared libraries were sequenced by HiSeq 2500 or NextSeq 500. The procedure for analyzing sequence data was previously described[3].

**Reversion analysis.** "Leave-one-out" strains were produced by P1 transduction, as described in the genomic reconstruction section. Each strain was constructed to contain four out of the five mutations of the minimally reconstructed set with one essential mutation reverted back to the wild-type allele (wild-type sequence in the locus). For the reversion analysis, these strains were taken from glycerol stocks and transformed with the pCBB plasmid. Transformations were plated on M9 minimal media supplemented with pyruvate 0.5%, glycerol 0.1% (W/V), and 34 mg/l chloramphenicol. Notably, under these permissive conditions, hemiautotrophic growth and sugar synthesis from $CO_2$ are not selected for as all sugar-derived

biomass building blocks can be synthesized from the supplied glycerol. Several colonies for each strain were inoculated (≈$10^4$ cells/inoculum) in 5 ml of permissive media and incubated in a high (10%) $CO_2$ incubator. In late exponential phase (≈1–2 OD), the cells were washed once with M9 minimal media and plated on large (15 cm) M9 minimal media agar plates supplemented with 0.5% pyruvate and 34 mg/l chloramphenicol. Under these conditions, growth is dependent on the activity of the CBB cycle for the production of all sugar-derived biomass building blocks from $CO_2$[3], (hemiautotrophic growth). Following 1–2 weeks of incubation, colonies were inoculated in restrictive media to further validate the hemiautotrophic growth in liquid. Leave-one-out strains capable of hemiautotrophic growth were sequenced to identify new mutations. For each leave-one-out strain, we plated independent cultures. Multiple colonies were sequenced from each plate. Sequencing was initially done by Sanger sequencing of the loci that corresponds to the reverted mutation in each strain (e.g., *prs* in the *prs*WT leave-one-out strain). If a mutation was not detected by Sanger sequencing, we performed whole-genome sequencing. Mutational analysis was done by comparing the genomic variations before and after the reversion analysis.

In the case of ppsRWT strains that managed to grow on pyruvate plates, we inoculated many colonies in 0.5% liquid pyruvate supplemented with 34 mg/l chloramphenicol. If growth was detected, we diluted the culture at a ratio of 1:100 or 1:1000, and sequenced the *ppsR* locus. We found that all cultures showed a mutation in *ppsR* that accumulated during the liquid growth phase.

**Protein expression and in vitro activity assay for SerA.** Expression protocol: serA (D-3-phosphoglycerate dehydrogenase) mutants and wild-type coding DNA sequence (CDS) were cloned into a pET-28a plasmid (Novagen) modified with a strep tag (KanR). After cloning and verification, the plasmids were transformed into an *E. coli* BL21star/DE3 (Invitrogen) strain (capR) for expression. Starters were placed overnight in LB + Kan 50 μg/ml + CAP 30 μg/ml (2 ml) and grow at 37 C. The cultures were diluted at a ratio of 1:100 to 100 ml of LB + Kan (50 μg/ml) and grown until an OD of 0.5–0.6. IPTG was added to a final concentration of 0.5 mM and incubated at 20° C overnight. The cultures were harvested and frozen until further use (−20 °C).

Purification protocol: The frozen pellets were resuspended in 30 ml of Buffer W (100 mM Tris/HCl, pH 8.0, 150 mM NaCl, and 1 mM EDTA) supplemented with protease inhibitor cocktail (Sigma, Cat No. P-2714). Cells were lysed by sonication (10 times for 30 s at 30% amplitude with intervals of 60 s in ice). The lysed cells were centrifuged for 45 min at 15,000 rpm (38,000 xg), at 4° C, and the supernatant was filtered using a 0.45-μm filter. Preparation of columns for purification was performed according to the following procedure: a) 2 ml of strep-tag resin (Strep-Tactin Sepharose 50% solution—Iba Technologies) was loaded and packed onto a gravity flow column, and b) the resin was washed with 10 volumes of Buffer W. Following the columns preparation, the filtered supernatant was loaded. The column was washed twice with 10 volumes of Buffer W and eluted with 5 volumes of Buffer E (Buffer W plus 2.5 mM desthiobiotin). Dialysis and concentration were done by ultracentrifugation following the manufacturer's condition (Centricon tubes, 10000-KDa cutoff, Millipore) into final buffer (100 mM Tris/HCl, pH 7.5, 150 mM NaCl, and 1 mM DTT). Enzymes were stored at 4° C at a concentration of 1–2 mg/ml.

In vitro activity assay: The enzymes were assayed for in vitro activity by using 12.6 μg/ml of purified protein in 50 mM phosphate buffer, pH 7.5, and supplying 0.8 mM NADH and 1 mM oxoglutarate. The kinetics of oxoglutarate reduction were quantified by following the decrease in absorbance at 340 nm. A total of 96-well plates are suitable for the UV range (Microplate UV/VIS 96 F, Eppendorf) and 200 μl reaction volumes were used throughout.

**In vivo assay for testing serA mutants.** A *serA* deletion was introduced into BL21star/DE3 (Invitrogen) strain (capR) using P1 transduction, as described in the genomic reconstruction section (JW2880), to create a Δ*serA* strain. pET-28a plasmids containing wild-type or serA mutant alleles (used for the in vitro assay) were transformed to Δ*serA* strains and verified. For each tested strain, three independent colonies were inoculated into M9 supplemented with glucose 0.5%, chloramphenicol 34 mg/l, and a variable level of IPTG (0 mM, 0.005 mM, 0.01 mM, 0.02 mM, 0.04 mM, 0.08 mM, and 0.16 mM). Reported growth rates are the average of three biological replicates. At the end of each experiment, each variant was sequenced with pET F/R generic primers to ensure that no reversion mutants were observed.

**Directed proteomics for testing PpsA state.** Sample preparation: 1-ml culture (1 ml) of ancestor strain was grown on M9 supplemented with pyruvate 0.5% and xylose 0.02%. 1 ml culture of evolved strains (Ex. 1, Ex. 2, and Ex. 4 from ref.[3]) was grown on M9 supplemented with pyruvate 0.5%. Three biological repeats of each strain were grown to mid-log in elevated (10%) $CO_2$ for harvesting.

The PpsA peptide, TCHAAIIAR, containing either an unmodified or phosphorylated threonine residue was purchased from JPT Peptide Technologies GmbH (Berlin, Germany). All chemicals were purchased from Sigma-Aldrich, unless stated otherwise. Samples were subjected to in-solution tryptic digestion using a modified Filter-Aided Sample Preparation protocol[32] (FASP). Sodium dodecyl sulfate buffer (SDT) included 4% (w/v) SDS, 100 mM Tris/HCl, pH 7.6, and 0.1 M DTT. Urea buffer (UB): 8 M urea (Sigma, U5128) in 0.1 M Tris/HCl, pH

8.0 and 50 mM ammonium bicarbonate. Cells were dissolved in 100 µl of SDT buffer and lysed for 3 min at 95 °C. Then, they were centrifuged at 16,000 X $g$ for 10 min. 100 µg of total protein were mixed with 200 µl of UB, loaded onto a 30 kDa molecular weight cutoff filter, and centrifuged. 200 µl of UB were added to the filter unit and centrifuged at 14,000 x g for 40 min. Proteins were alkylated using 100 µl of iodoacetamide (IAA), and washed twice with ammonium bicarbonate. Trypsin was then added along with an aliquot of the synthetic peptide mix (equal volume for all samples), and samples were incubated at 37 °C overnight. Additional trypsin was added and incubated for 4 h at 37 °C. Digested proteins were then centrifuged into a clean collection tube. 50 µl of NaCl at a concentration of 0.5 M was added to the filter, centrifuged, acidified with trifluoroacetic acid, desalted using HBL Oasis, speedvaced to dryness, and stored at −80 °C until analysis.

Liquid chromatography: ULC/MS-grade solvents were used for all chromatographic steps. Each sample was loaded using splitless nano-Ultra-Performance Liquid Chromatography (10-kpsi nanoAcquity; Waters, Milford, MA, USA). The mobile phase was A) $H_2O$ + 0.1% formic acid and B) acetonitrile + 0.1% formic acid. Desalting of the samples was performed online using a reversed-phase C18 trapping column (180-µm internal diameter, 20-mm length, and 5-µm particle size; Waters). The peptides were then separated using a T3 HSS nanocolumn (75-µm internal diameter, 250-mm length, and 1.8 µm-particle size; Waters) at 0.35 µl/min. Peptides were eluted from the column into the mass spectrometer using the following gradient: 4–30%B in 105 min, 30–90%B in 5 min, maintained at 90%B for 5 min, and then back to initial conditions.

Mass spectrometry: The nanoUPLC was coupled online through a nanoESI emitter (10-µm tip; New Objective; Woburn, MA, USA) to a quadrupole orbitrap mass spectrometer (Q Exactive Plus, Thermo Scientific) using a FlexIon nanospray apparatus (Proxeon).

For "Targeted" analysis, data were acquired in parallel reaction monitoring (PRM) mode, where MS2 resolution was set to 35,000 and the maximum injection time was set to 100 msec. The inclusion list included all relevant peptides in the isotopically labeled or unlabeled form.

For calculating the total ion current (TIC), which was used for loaded peptide amount normalization, data were acquired in data-dependent acquisition (DDA) mode, using a Top20 method. MS1 resolution was set to 70,000 (at 400 m/z) and maximum injection time was set to 20 msec. MS2 resolution was set to 17,500 and maximum injection time of 60 msec.

Raw data were imported into the Skyline software[33]. Each peptide was manually curated to select the most intense and reliable transitions, as well as to determine the exact peak boundaries. Peak areas were then exported to a Microsoft Excel file, where quantitative comparisons based on TIC-normalized peak areas were performed. A Student's t-test, after logarithmic transformation, was used to identify significant differences across the biological replica. Fold changes were calculated based on the ratio of arithmetic means of the case (evolved strain) vs. control (ancestor) samples.

An additional phosphorylation site in the PpsA proteins is located in a histidine residue adjacent to the regulatory threonine. The phosphorylation of this residue is catalytic in nature[6]. As both our standard peptides (containing an unmodified or phosphorylated threonine residue) have an unmodified histidine residue, our targeted proteomic analysis does not include copies of the protein which are phosphorylated in both histidine and threonine residues.

**Isotopic analysis of hydrolyzed amino acids**. Isotopic analyses are described in detail in ref. [3]. For mass isotopic distribution analysis, cells were cultured in 5 ml of M9 minimal media supplemented with 0.2% (w/v) non-labeled sodium pyruvate (Sigma-Aldrich, USA). Cultures were placed in transparent, airtight, 250 ml Erlenmeyer flasks sealed with a rubber septum, and the headspace of the flasks was flushed with ≈5 volumes of isotopically labeled $^{13}CO_2$ gas mixture (Cambridge Isotope Laboratories, USA) composed of 10%$^{13}CO_2$, 10% $O_2$, and 80% $N_2$. The sealed flasks were placed in a shaking incubator (37 °C, 220 RPM) for 24–48 h. Following the incubation period, the flasks were opened, and 1 ml of saturated culture was harvested by centrifugation at 10,000xg. The cell pellet was washed twice with M9, resuspended in 1 ml of 6 N hydrochloric acid, and incubated for 24 h at 110 °C. The hydrochloric acid was evaporated under nitrogen stream for ~ 1 h at 95 °C. The resulting residue was resuspended in 750 µl of ultrapure water and vortexed thoroughly. The suspension was then centrifuged for 5 min at 10,000 x $g$. 200 µl of the supernatant were transferred into wells of a microplate and 5 µl were injected to an ultra-performance liquid chromatography (Acquity—Waters, USA) on a C-8 column (Zorbax Eclipse XBD—Agilent, USA) at a flow rate of 0.6 ml/min and eluted off the column using a hydrophobicity gradient. Buffers used were A) $H_2O$ + 0.1% formic acid and B) acetonitrile + 0.1% formic acid with the following gradient: 100% of A (0–3 min), 100% A to 100% B (3–9 min), 100% B (9–13 min), 100% B to 100% A (13–14 min), and 100% A (14–20 min). The overall run time was 20 min. The UPLC was coupled online to a triple-quadrupole mass spectrometer (TQS—Waters, USA). Data were acquired using MassLynx v4.1 (Waters, USA). Optimization of ionization parameters and determination of retention times was performed by direct infusion of amino acid commercial standards (Sigma-Aldrich, USA). Argon was used as the collision gas with a flow rate of 0.22 ml/min. Cone voltage was 25 V, the capillary was set to 3 kV, source temperature was 150 °C, desolvation temperature was 500 °C, desolvation gas flow was 700 L/min, source offset 50, cone gas flow was 250 L/min, and collision energy was 14 eV. MIDs were detected using multiple-reaction monitoring (MRM) with the known molecular masses and the neutral loss of carbonyl carbon as a daughter ion (either 47 or 46 m/z, labeled and unlabeled, respectively). Data analysis was performed using TargetLynx (Waters, USA).

**Data availability**. Whole-genome sequencing data that support the findings of this study have been deposited at The European Nucleotide Archive (ENA) under PRJEB21762 (ERP024048) study. All other relevant data are available from the corresponding author on request.

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

## Acknowledgements

We thank Elad Noor, Arren Bar Even, Avi Flamholz, Gal Ofir, Gil Amitai, Rotem Sorek, Avigdor Eldar, Ahuva Cooperstein, Shaked Cahanovitc, Elad Itzhaki, Amit Bachar, Sagit Yahav, Lior Zelcbuch, Adar Lopez, Ghil Jona, and the bacteriology team, Shlomit Gilad, Sima Benjamin, and the INCPM genomics team, Yishai Levin, Dan Yakir, Roy Kishony, Idan Yelin, Dan Tawfik, and Tomer Shlomi. This work was funded by the European Research Council (projects SYMPAC 260392 and NOVCARBFIX 646827), Israel Science Foundation 740/16, Dana and Yossie Hollander, the Helmsley Charitable Foundation, the Larson Charitable Foundation, the Estate of David Arthur Barton, the Anthony Stalbow Charitable Trust, and Stella Gelerman, Canada. R.M. is the Charles and Louise Gartner professional chair. D.G.W. is supported by the United States–Israel Education Foundation.

## Author contributions

E.H., N.A., and R.M. designed the project. E.H., K.L.F., Y.Z., and L.N.-G. conducted the experiments. E.H., N.A., and Y.B.O. analyzed the sequencing data. A.S. and S.G. performed and analyzed the proteomics assay. E.H., N.A., U.B., D.D., and Y.B.O. performed the computational analysis. E.H and D.G.W performed plasmid engineering. E.H., S.G., and N.A. performed and analyzed the LC/GS–MS assays. E.H., N.A., Y.B.O., N.P., and RM wrote the paper.

## Additional information

**Competing interests:** The authors declare no competing financial interests.

