## [Peer Review File · Nature Communications]

Reviewers' comments:

Reviewer #1 (Remarks to the Author):

Herz, Antonovsky et al. follow up on their recent report of an E. coli strain that is able to assimilate CO₂ into sugars via a synthetically constructed 'hemiautotrophic' setup in which central metabolism is split into two modules. They are (impressively) able to reconstruct a minimal set of five mutations that are able to confer this phenotype in the context of their designed metabolism. Then, they perform a "leave-one-out suppressor screen" to characterize the full variety of mutations that could substitute in for each of the five key mutations. Finally, they biochemically characterize how the mutations that they find in three of these enzymes affect their activities. This is a well-designed and executed study that provide unusually detailed information about how bacterial metabolism can adapt in response to new constraints and opportunities.

Comments

1. In the abstract, some minor wordsmithing is needed. First, the language should be less absolute: "are masked by" would be better as "may be masked by". Also "enable robust growth where" should be "when". Lastly, for "highlight the contribution of carbon balance in autocatalytic cycles" it needs to contribute to something. Maybe: "highlight the contribution of proper carbon balance to optimizing metabolic pathways that incorporate autocatalytic cycles".
2. In the intro, minor changes: It's probably not true that "each strain contained 20 to 40 "fine-tuning" mutations". As it says in the next paragraph, some of these were probably near-neutral mutations that accumulated due to hitchhiking or genetic drift. Please clarify the phrasing here. This phrase "the molecular basis that failed the original design" should be something more like "the molecular basis of shortcomings in the original design".
3. For the P1 transduction from Keio strains, two details are missing and should be provided in the methods or a supplement: (1) Which Keio knockout was used as a donor for transduction? (2) Was the kanamycin resistance cassette that is present in the Keio strain and used for selection removed via FLP recombination in the final strains that were tested? The methods refer to Table S6 for the Keio strains used, but it seems to only list 4 of the 5 alleles that were reverted in constructing the minimal set, and I believe that there were 9 alleles in Table S1 that were reverted in this fashion. The image in Figure S1 is helpful for the reconstruction protocol, but it doesn't make clear whether there is still the KanR cassette in the chromosomes of the final strains.
4. There needs to be more explanation or a table to explain the nomenclature for some of the more complex mutations like this one (e.g., prs:R105_A110dup).
5. For this statement: "More detailed investigation of the growth of these strains revealed that they can grow hemiautotrophically in rare cases, always with much slower growth rate and low final yield". Presumably the authors can't rule out that these instances of growth are due to the rapid appearance of additional mutations as they mention later. It may be worth noting that this is the most likely explanation for the sporadic cases of growth.
6. Fig. 2. " Ellipses mark the feed substrates." I don't see any ellipses. The substrates seem to be in ovals.
7. The connection between the statement "deep sequencing and for each of the evolved mutations independently" in the results and the sequencing of colonies in the Fig. 4 caption is not very clear. It seems to indicate that the authors picked many suppressor mutations and sequenced them together as a pool, but it looks like they just picked multiple colonies and sequenced each one separately. Were the colonies all derived from one culture and one plate for each leave-one-out strain or did they do separate cultures as in a fluctuation test to ensure independent mutations in

each colony that they included in sequencing? None of these details impact the results – they saw different mutations and the diversity of those is what is important, but the methods need to be clearer.

8. In Fig. 4, it may be difficult (esp. for the color blind) to tell the difference between the three stars with different colors. I'd recommend using a different symbol (in addition to a different color) for each category of mutation to make this visually clearer.

9. Regarding, " Due to the natural error rate during DNA replication, such a large population will contain all possible point mutants in the E. coli genome", this is the expectation based on a back-of-the-envelope calculation, but mutational biases and sampling error mean that it's probably not entirely true. Maybe change to "is likely to contain nearly all possible point mutants".

10. "A shared design principle of the mutations related to flux branch points out of the CBB cycle is reduction of the catalytic capacity of the flux branch point reactions." Is this statement referring to a reduction in the reaction in the branch point that is out/in of the catalytic cycle or of the step within the catalytic cycle at the branch? It would be worth expanding on the explanation of the theory here since it is a central point.

11. Comment: It may be worth mentioning that even though all mutations seem to result in less functional versions of the mutated genes, at least one is a loss-of-function mutation in a negative regulator, so it increases the activity of a key enzyme. When metabolism is regulated in this fashion (as it commonly is), it makes it far more likely to observe loss-of-function mutations in the negative regulator than gain-of-function mutations in an enzyme. It isn't that evolution can only "break" things, just that it's more likely to happen, so when it is an option, that option probabilistically dominates.

12. In the methods, I'm confused about why there is no tyrosine (Y) in this peptide sequence if it is the peptide that was profiled for phosphorylation by mass spec? -- "The PpsA peptide, TCHAAIIAR, containing either an unmodified or phosphorylated tyrosine residue".

Reviewer #2 (Remarks to the Author):

The manuscript of Herz & Antonovsky et al. builds on a recent report from the same lab, in which the authors described the successful conversion of E. coli from a heterotrophic into a "hemiautotrophic" organism by rational design and experimental evolution. In this manuscript, the authors describe efforts to rationalize the individual mutations that accumulated along the way of the experimental evolution experiments.

Although the manuscript is in general of high quality, well written in most parts and convincing in many aspects, the authors only live partially up to their promise to explain the observed mutations. While the authors are easily satisfied with explanations that fit their hypothesis on individual mutations (e.g. SerA, pgi), they miss the chance to explain the actual interesting and non-expected mutations they observe.

Most notably, the fact that prs mutation becomes suddenly non-essential stands in strong contrast to the conclusions drawn by the very same authors in their first publication, where they clearly state that this mutation is a necessary requirement, because it is an essential branching point (Antonovsky et al. Cell 166:115-125, pp. 120). For this particular gene it is not appropriate simply to conclude that other local minima were hit by the multiple suppressor mutations. I expect a detailed analysis on the effect of the individual suppressor mutations detected in the experiment. The fact that multiple mutations can compensate the prs mutation most probably shows that this gene is not essential at all to establish hemiautotrophy. This is the most important finding of the

whole manuscript that is not addressed at all!

Other, minor points:

P1, L15, P18, L9 and elsewhere: catalytic capacity. Please introduce the concept to the reader. Is catalytic capacity a k_{cat} or k_{cat}/K_M argument, or depending on the metabolic context even both? Please take the time to discuss this concept a bit more in detail!

P2, L2: The CBB does NOT account for more than 99% of the global carbon cycle. Please correct this statement or weaken it (e.g., more than 90%, overwhelming majority,...).

P3, L5: "augmented" strain – in which respect "augmented"? Is the strain not rather "prepared" for the experimental evolution (or even "weakened" in respect to the normal physiological function). Please use other, more explanatory, objective wording, or even better do not use any adjectives and simply state "mutant strain"?

P3, L15: "making", not the right word in this context. Designing or reprogramming would be more appropriate, or even better: evolving, as most of the success comes from experimental evolution, not from rational design.

P6, L4: "it WAS not require", verb is missing

P6, L6: "rare cases" Ambiguous. Please be more specific here. It is unclear to the reader "which strains" you actually mean (i.e. the ppsR) and it is completely unclear what "rare cases" mean (that only in some cases the transfer into liquid culture ended up in growth and not that specific conditions were required to allow for growth). Rephrase the sentence to be more explicit.

P14, L18: This experiment is not sufficient to demonstrate that the enzyme activity is affected, because the mutations could also touch the substrate specificity and thus only the activity for the substrate mimic. Can you please place a caveat here (I agree that together with the experiment below, the explanation given is likely, but I would nevertheless appreciate a caveat here)!

P15, L3: The factor 2 in growth rate is not reflecting the factor 10-100 measured with the enzymes. The higher expression level (expression from the plasmid) might compensate activity loss and enzyme activity might not be limiting, completely. Can you put this caveat here?

We thank the reviewers and you for the careful reading of our paper and the thoughtful comments. We have updated the text to address the issues raised by the referees. Below we give detailed point by point responses that answer the reviewers' comments and enhance the quality of the paper.

Reviewer #1 (Remarks to the Author):

Herz, Antonovsky et al. follow up on their recent report of an E. coli strain that is able to assimilate CO₂ into sugars via a synthetically constructed 'hemiautotrophic' setup in which central metabolism is split into two modules. They are (impressively) able to reconstruct a minimal set of five mutations that are able to confer this phenotype in the context of their designed metabolism. Then, they perform a "leave-one-out suppressor screen" to characterize the full variety of mutations that could substitute in for each of the five key mutations. Finally, they biochemically characterize how the mutations that they find in three of these enzymes affect their activities. This is a well-designed and executed study that provide unusually detailed information about how bacterial metabolism can adapt in response to new constraints and opportunities.

Comments

1. In the abstract, some minor wordsmithing is needed. First, the language should be less absolute: "are masked by" would be better as "may be masked by". Also "enable robust growth where" should be "when". Lastly, for "highlight the contribution of carbon balance in autocatalytic cycles" it needs to contribute to something. Maybe: "highlight the contribution of proper carbon balance to optimizing metabolic pathways that incorporate autocatalytic cycles".

We thank Reviewer#1 for the thoughtful and detailed feedback. In the revised manuscript the abstract has been modified in-line with the reviewer's suggestions.

2. In the intro, minor changes: It's probably not true that "each strain contained 20 to 40 "fine-tuning" mutations". As it says in the next paragraph, some of these were probably near-neutral mutations that accumulated due to hitchhiking or genetic drift. Please clarify the phrasing here. This phrase "the molecular basis that failed the original design" should be something more like "the molecular basis of shortcomings in the original design".

We modified the introduction according to the reviewer's comments. We acknowledge that only a small fraction of the mutations found are phenotypically relevant and hence eliminated "fine-tuning" from the sentence. In addition we have re-phrased the concluding sentence in the introduction in-line with the reviewer's suggestion.

3. For the P1 transduction from Keio strains, two details are missing and should be provided in the methods or a supplement: (1) Which Keio knockout was used as a donor for transduction? (2) Was the kanamycin resistance cassette that is present in the Keio strain and used for selection removed via FLP recombination in the final strains that were tested? The methods refer to Table S6 for the Keio strains used, but it seems to only list 4 of the 5 alleles that were reverted in constructing the minimal set, and I believe that there were 9 alleles in Table S1 that were reverted in this fashion. The image in Figure S1 is helpful for the reconstruction protocol, but it doesn't make clear whether there is still the KanR cassette in the chromosomes of the final strains.

(1) We have modified Table S6 to now contain the KEIO strain identifier explicitly. As noted, 4 out of the 5 mutations have been transduced using an adjacent marker (ychH/dauA for prs, yjbl for pgi,yqfA for serA, chiA for crp). In the case of ppsR, the mutations found in the evolved hemiautotrophic strains were loss-of-function

mutations. As such, to introduce the ppsR mutation we transduced the ppsR:kana marker into the reconstructed strain using the ppsR (JW1693) KEIO strain as the donor. In the revised manuscript we are now noting this explicitly in the relevant methods section.

(2) We thank the reviewer for highlight the ambiguity regarding the KanR cassette arising from figure S1. The resistance cassette has not been removed in the final strain but only during the iterative process to allow following iteration to take place. In the revised manuscript we now state that the KanR cassette have not been removed in the final strain in both Method section and the caption of Figure S1.

4. There needs to be more explanation or a table to explain the nomenclature for some of the more complex mutations like this one (e.g., prs:R105_A110dup).

While we attempted to follow standard nomenclature for describing the mutations we acknowledge that further description in the text is required. In the revised manuscript we added clarification regarding non-trivial mutations where such are first introduced in the text. In addition we revised the “notes” column in the the supplementary tables which specify the mutations found in the suppressor analysis is now contain an explanatory note regarding non-trivial mutations.

5. For this statement: "More detailed investigation of the growth of these strains revealed that they can grow hemiautotrophically in rare cases, always with much slower growth rate and low final yield". Presumably the authors can't rule out that these instances of growth are due to the rapid appearance of additional mutations as they mention later. It may be worth noting that this is the most likely explanation for the sporadic cases of growth.

In the revised manuscript we further detail the results obtained regarding the ppsR “leave-one-out” cultures which exhibit slow and sporadic growth. Specifically, we have sequenced some of these cultures following the slow growth phase. We were unable to detect additional mutations in these strains. However, when these cultures are further propagated (for example, by dilution into fresh media), ppsR mutants rapidly arise. In the revised manuscript we describe these findings and speculate regarding the source of the observed growth in unmutated clones. As detailed below, it is possible that some genetic mutations could not be detected in our usage of next generation sequencing (for example, some forms of chromosomal rearrangements which are not detected in the utilized pipeline). We have included the modified paragraph in the Results section:

“However, when colonies were inoculated in liquid media, most inoculums did not exhibit growth. About 10% of cases, usually with a large inoculum, showed some slow growth with very low yield (Fig S7). In these cultures, whole genome sequencing did not reveal additional mutations that could explain the observed growth. We speculate that the sporadic growth (in the absence of additional mutations) can result from a stochastic transition between metabolic states that occurs only in a small subfraction of the population (Kotte, Oliver, et al. Molecular systems biology 2014) or due to a mutation we could not detect in our usage of next generation sequencing.”

6. Fig. 2. " Ellipses mark the feed substrates." I don't see any ellipses. The substrates seem to be in ovals.

We thank the reviewer for the geometric insight and modified the text accordingly.

7. The connection between the statement "deep sequencing and for each of the evolved mutations independently" in the results and the sequencing of colonies in the Fig. 4 caption is not very clear. It seems to indicate that the authors picked many suppressor mutations and sequenced them together as a pool, but it looks like they just picked multiple colonies and sequenced each one separately. Were the colonies all derived from one culture and one plate for each leave-one-out strain or did they do separate cultures as in a fluctuation test to ensure independent mutations in each colony that they included in sequencing? None of these details impact the results – they saw different mutations and the diversity of those is what is important, but the methods need to be clearer.

We thank the reviewer for highlighting the ambiguity in the experimental description. For each leave-one-out strain we inoculated several independent cultures ($n > 4$) which were then plated on distinct agar plates (to ensure independent mutations, as in fluctuation test). From each plate we picked and sequenced multiple colonies. As expected, most of the clones picked from the same plate contained identical mutations however in some cases distinct mutations arose in a single culture. For the sake of graphical simplicity we present a single plate from which multiple mutations were identified. However, in the figure itself, only unique mutations are presented (meaning each marker corresponds to a distinct genotype and genotypes are specified in the supplementary tables). To clarify the experimental procedure we have modified the text in the results and methods section, as well as as the caption of figure 4.

8. In Fig. 4, it may be difficult (esp. for the color blind) to tell the difference between the three stars with different colors. I'd recommend using a different symbol (in addition to a different color) for each category of mutation to make this visually clearer.

We have modified the symbols accordingly and further verified with our "in-house" color blind scientist that the difference between mutations types is noticeable.

9. Regarding, "Due to the natural error rate during DNA replication, such a large population will contain all possible point mutants in the E. coli genome", this is the expectation based on a back-of-the-envelope calculation, but mutational biases and sampling error mean that it's probably not entirely true. Maybe change to "is likely to contain nearly all possible point mutants".

In the revised manuscript the sentence has been modified in-line with the reviewer's suggestions.

10. "A shared design principle of the mutations related to flux branch points out of the CBB cycle is reduction of the catalytic capacity of the flux branch point reactions." Is this statement referring to a reduction in the reaction in the branch point that is out/in of the catalytic cycle or of the step within the catalytic cycle at the branch? It would be worth expanding on the explanation of the theory here since it is a central point.

In the revised manuscript we have added a sentence that aims to clarify the ambiguity regarding the influx/efflux directionality of catalytically reduced reaction in the results section:

"In all three biomass sinking branch points (*pgi*, *serA*, *prs*), in which intermediates are drained from the CBB cycle towards biomass synthesis, we observe mutations that are expected to decrease the catalytic capacity of the efflux reaction."

In addition we extended our referral to the theoretical prediction regarding the kinetic constraints of branching points in autocatalytic cycles:

"This is in line with theoretical predictions based on stability analysis of autocatalytic pathways like the CBB cycle where kinetic properties of enzymes at metabolic branch points were found to determine whether the cycle can operate stably^{4,14}."

Due to the limited length of the manuscript, we prefer to refer the curious reader to the In-depth mathematical analysis, as well as underlying metabolic intuition, which have been extensively discussed in recent publications (Barenholz et al. eLIFE 2017, Antonovsky et al. Cell 2016) to which references are supplied.

11. Comment: It may be worth mentioning that even though all mutations seem to result in less functional versions of the mutated genes, at least one is a loss-of-function mutation in a negative regulator, so it increases the activity of a key enzyme. When metabolism is regulated in this fashion (as it commonly is), it makes it far more likely to observe loss-of-function mutations in the negative regulator than gain-of-function mutations in an enzyme. It isn't that evolution can only "break" things, just that its more likely to happen, so when it is an option, that option probabilistically dominates.

We like the insight brought by the reviewer and agree with the logic presented. In trying to incorporate this into the text we found that because this is a single case in our experiment, it feels like possibly reading too much into the data we have at hand. We thus prefer not to explicitly speculate on this matter and hopefully in future studies we or other labs can come back to this issue.

12. In the methods, I'm confused about why there is no tyrosine (Y) in this peptide sequence if it is the peptide that was profiled for phosphorylation by mass spec? -- "The PpsA peptide, TCHAAIIAR, containing either an unmodified or phosphorylated tyrosine residue".

We thank the reviewer for noticing our mistake regarding the phosphorylation site in ppsA. The phosphorylated amino acids is obviously threonine (T) and not tyrosine (Y) as wrongly written in the text. In the revised manuscript we corrected this mistake.

Reviewer #2 (Remarks to the Author):

The manuscript of Herz & Antonovsky et al. builds on a recent report from the same lab, in which the authors described the successful conversion of E. coli from a heterotrophic into a "hemiautotrophic" organism by rational design and experimental evolution. In this manuscript, the authors describe efforts to rationalize the individual mutations that accumulated along the way of the experimental evolution experiments.

Although the manuscript is in general of high quality, well written in most parts and convincing in many aspects, the authors only live partially up to their promise to explain the observed mutations. While the authors are easily satisfied with explanations that fit their hypothesis on individual mutations (e.g. SerA, pgi), they miss the chance to explain the actual interesting and unexpected mutations they observe.

Most notably, the fact that prs mutation becomes suddenly nonessential stands in strong contrast to the conclusions drawn by the very same authors in their first publication, where they clearly state that this mutation is a necessary requirement, because it is an essential branching point (Antonovsky et al. Cell 166:115-125, pp. 120). For this particular gene it is not appropriate simply to conclude that other local minima were hit by the multiple suppressor mutations. I expect a detailed analysis on the effect of the individual suppressor mutations detected in the experiment. The fact that multiple mutations can compensate the prs mutation most probably shows that this gene is not essential at all to establish hemiautotrophy. This is the most important finding of the whole manuscript that is not addressed at all!

As indicated by reviewer#2, in our recent publication (Antonovsky et al. Cell 166:115-125, pp. 120) we tested the essentiality of the prs mutation (that emerged in three independent chemostat experiments) for the hemiautotrophic phenotype and indeed found it was essential in all three cases. In addition, we presented a simplified kinetic model which suggested that a stable, non-zero steady state flux exists only when the kinetic

parameters of enzymes at the flux branch point is within derived constraints. Recently, we have extended the theoretical analysis on the stability of autocatalytic pathways beyond the simplified model presented in Antonovsky et al. Cell 2016 to include scenarios in which more than one branch point in the cycle exists. As reported in (Barenholz et al. eLife 2017), when more than one branch point exists the stability criterion is a function of the kinetic parameters (K_m , K_{cat}) of all branch points enzymes. As such, stable metabolic activity can be achieved by changes in one or more of the enzymes.

In the current manuscript we analyze *prs* “leave-one-out” strain which is not mutated in *prs* but contains two additional mutations in flux branch points (*pgi* and *serA*). This is in contrast to the strains that were tested for *prs* essentiality in (Antonovsky et al. Cell 166:115-125, pp. 120), in which *prs* was the only branch point reaction in which mutations were found (excluding the *glmU* mutation found in exp. 1, that affects a F6P shunting towards LPS synthesis. However, this is the smallest branch point in terms of carbon mass requirements in the cycle <1%). Hence, in the genetic background of the *prs* “leave-one-out” strain, our model suggests that stable operation of autocatalytic cycle can potentially be achieved in the absence of mutations in *prs*.

Yet the question arises, why then would additional mutations (e.g., *pitA*, *proA*, *cpxA*, *gpp*) permit hemiautotrophic growth in the *prs* leave-one-out strain. While we are baffled by this result, a better experimental resolution will be needed to further investigate this issue and suggest a role for these alternative mutations.

We acknowledge that additional reference to this result is required and in the revised manuscript the following paragraph has been added to the results section:

"We speculate that following the introduction of the mutations in *serA* and *pgi*, which affect carbon branching from the CBB cycle, a mutation in *prs* is no longer required to ensure the autocatalytic stability of the cycle. This is in contrast to essentiality of the *prs* mutation in the genetic background lacking mutations in *serA* and *pgi*, as was previously observed³. It is not clear at this point how mutations that do not seem relevant to carbon balancing (e.g., *cpxA*, *proA*, *gpp*) can compensate to achieve hemiautotrophic growth in the *prs* leave-one-out background. Future work with better experimental resolution will be needed to further investigate this issue and suggest a role for these alternative mutations that compensate for the *prs* mutation in our leave-one-out test."

Other, minor points:

P1, L15, P18, L9 and elsewhere: catalytic capacity. Please introduce the concept to the reader. Is catalytic capacity a k_{cat} or k_{cat}/K_M argument, or depending on the metabolic context even both? Please take the time to discuss this concept a bit more in detail!

We thank the reviewer for highlighting the need to introduce the concept of catalytic capacity to the vast majority of readers. In the revised manuscript we have introduced a formal definition, as well as a brief introduction to the concept. In addition, we are now referring to a recent review which offers a detailed introduction of catalytic capacity concept and derived applications.

"The catalytic capacity of a reaction, often denoted as V_{max} , is defined as $E \cdot k_{cat}$, where E is the amount of the catalyzing enzyme and k_{cat} is the turnover rate of the enzyme¹³. Because k_{cat} is a constant, the catalytic capacity of an enzyme is proportional to the amount of enzyme expressed. As multiple enzymes are capable to shunt F6P towards biosynthesis pathways, the generalized catalytic capacity of the efflux reaction can be defined as the sum of the catalytic capacity of *Pgi* and the catalytic capacity resulting from promiscuous activity of other enzymes. Loss-of-function mutation in *pgi* is expected to tune-down the efflux capacity of the intermediate F6P from the CBB cycle towards G6P synthesis. This is consistent with other lines of evidence for the required fine tuning for the operation of the autocatalytic cycle¹⁴ as described in the discussion."

P2, L2: The CBB does NOT account for more than 99% of the global carbon cycle. Please correct this statement or weaken it (e.g., more than 90%, overwhelming majority,...).

In the revised manuscript we have modified the text with the reviewer's suggestion to "overwhelming majority".

Yet, as an aside we note that based on:

<http://bionumbers.hms.harvard.edu/bionumber.aspx?&id=109976&ver=1&trm=raven%20carbon>
which connects to Table 3 here:

Contributions of anoxygenic and oxygenic phototrophy and ...

www.int-res.com/abstracts/ame/v56/n2-3/p177-192/ by JA Raven - 2009

we were under the impression that it is indeed more than 99%.

We also note: Prokaryotes (2006) 2:32–85 DOI:10.1007/0-387-30742-7_3 which states:

"Today, the significance of anoxygenic photosynthesis for global carbon fixation is limited for two reasons. On the one hand, phototrophic sulfur bacteria (the dominant anoxygenic phototrophs in natural ecosystems) form dense accumulations only in certain lacustrine environments and in intertidal sandflats. The fraction of lakes and intertidal salt marshes that harbor anoxygenic phototrophic bacteria is unknown, but these ecosystems altogether contribute only 4 % to global primary production (Whittaker and Likens 1975). In those lakes harboring phototrophic sulfur bacteria, an average of 28.7 % of the primary production is anoxygenic (Overmann 1997). Consequently, the amount of CO₂ fixed by anoxygenic photosynthesis must contribute much less than 1 % to global primary production"

P3, L5: "augmented" strain – in which respect "augmented"? Is the strain not rather "prepared" for the experimental evolution (or even "weakened" in respect to the normal physiological function). Please use other, more explanatory, objective wording, or even better do not use any adjectives and simply state "mutant strain"?

We thank the reviewer for highlighting the problematic use of adjectives in this context. In the revised manuscript we refer to the ancestor strain into which additional mutations have been introduced as "modified ancestor".

P3, L15: "making", not the right word in this context. Designing or reprogramming would be more appropriate, or even better: evolving, as most of the success comes from experimental evolution, not from rational design.

In the revised manuscript we modified the phrasing in the sentence:

"Here we describe our success in identifying a compact subset of only five mutations (Fig. 1b) that are sufficient for the adaptation of a rationally designed ancestor *E. coli* to produce sugar from CO₂."

P6, L4: "it WAS not require", verb is missing

In the revised manuscript the text was corrected.

P6, L6: "rare cases" Ambiguous. Please be more specific here. It is unclear to the reader "which strains" you actually mean (i.e. the ppsR) and it is completely unclear what "rare cases" mean (that only in some cases the transfer into liquid culture ended up in growth and not that specific conditions were required to allow for growth). Rephrase the sentence to be more explicit.

In the revised manuscript we rephrased the paragraph in the results section :

"More detailed investigation of the ppsR "leave-one-out" strain (that contain the native *ppsR* sequence along the other four mutations) revealed that a small number of inoculated cultures (which depends on the inoculum size, and in our case was $\approx 10\%$) were able to grow hemiautotrophically, always with much slower growth rate and low final yield."

P14, L18: This experiment is not sufficient to demonstrate that the enzyme activity is affected, because the mutations could also touch the substrate specificity and thus only the activity for the substrate mimic. Can you please place a caveat here (I agree that together with the experiment below, the explanation given is likely, but I would nevertheless appreciate a caveat here)!

In the revised manuscript the sentence has been modified in-line with the reviewer's suggestion.

"As the decrease in the reaction rate measured for the non-native substrate does not directly indicate a decrease in the reaction rate of the native substrate (for example when the mutation affects substrate specificity), we conducted an additional experiment to investigate the effect of the mutations on the native activity of the enzyme *in vivo*."

P15, L3: The factor 2 in growth rate is not reflecting the factor 10-100 measured with the enzymes. The higher expression level (expression from the plasmid) might compensate activity loss and enzyme activity might not be limiting, completely. Can you put this caveat here?

In the results presented in Fig. S5, an increase in the concentration of the inducer results in enhanced growth rate (even for the wild-type *serA* sequence), and hence we conclude that *serA* activity limits growth rate at low induction levels in the $\Delta serA$ strain. As suggested by the reviewer we now acknowledge in the text that the observed decrease in growth rate between native and mutant *serA* strains is significantly lower than the expected effect predicted by the *in vitro* measurements (for example, at low inductions levels). As suggested by the reviewer this discrepancy may stem from the difference in catalysis between the native (3PG) and the mimic substrate (2-oxoglutarate) in the mutant enzyme.

REVIEWERS' COMMENTS:

Reviewer #2 (Remarks to the Author):

The authors addressed most of my concerns and clarified most of the open questions to my satisfaction. I appreciate the authors' efforts of rephrasing the text to avoid misunderstandings and to provide more information on experimental procedures and the conclusions drawn.